# Impact Analysis of Regional Smart Development on the Risk of Poverty among the Elderly

**Chunyang Luo, Hongmei Li * and Lisha Song**

College of Science, North China University of Technology, Beijing 100144, China; 18800158862@163.com (C.L.); lssong@ncut.edu.cn (L.S.)

* Correspondence: lhmei6280@163.com

**Abstract:** As China continues to introduce policies to promote the construction of smart cities, the governance capacity and living environment of many pilot regions have moved towards smart development and sustainability. In order to reveal the impact of improving regional smart development on the lives of the elderly, we explored the relationship between regional smart development and the risk of poverty in old age. The results show that at present, the development of smart cities continues to grow; the majority of elderly respondents' poverty risk is general; the development of smart cities at the regional level is effective in reducing the poverty risk of the elderly in the region, with the degree of impact varying by region; and the impact of smart development at the regional level on the risk of poverty in old age varies with different levels of social support. Based on this, suggestions are made to vigorously develop the regional economy, improve the social security mechanism for the elderly, and accelerate the digitalization and humanization of infrastructure so as to better meet the needs of China's elderly population in the context of high-quality smart development in the region and achieve sustainable development goals.

**Keywords:** regional smart development; risk of poverty in old age; evaluation index system

## 1. Introduction

Regional smart development plays an important role in the integration of informationization, industrialization, urbanization construction, and sustainable development, etc. However, the development of regional intelligence relies on the Internet and emerging technology, which requires a high degree of acceptance of new things and learning ability, and most of the elderly population under this rapid development mode have not formed the digital literacy and thinking cognition to match; thus, the development of regional intelligence and smart cities does not have a significant impact on the happiness of this group [1].

The risk of poverty in old age refers to the greater vulnerability of the elderly to poverty due to changes in a variety of factors, such as physical characteristics, family status, and social roles [2]. Specifically in the process of social digitalization, the elderly population is less able than other groups to adapt to changes in their environment, lifestyles, and policy directions, leading to the increasingly prominent problem of returning to poverty due to old age and illness, as well as the problem of incapacitation [3]. By the end of 2020, the domestic elderly population reached 190 million people, which was up by 4.6% compared with 2010. The aging problem is becoming more serious. Although, during this period, China has eliminated absolute poverty through precise poverty alleviation (precise poverty alleviation is a poverty alleviation method that accurately identifies and helps to meet poverty alleviation objectives in different poverty situations), the old age allowance (the old age allowance (OAA) is a social security system proposed to solve the basic living problems of the elderly), and other measures, compared with the rapid development of aging, China's economic foundation for coping with the aging of the

population is still relatively weak. The risk of poverty in old age remains a very serious problem. Therefore, the present study will focus on how the rapid development of smart cities at the regional level specifically affects the risk of poverty in old age and will explore the different impacts of the development of smart cities on the risk of poverty in old age from the perspectives of different regions, different intergenerational family relationships, and different individual characteristics.

## 2. Literature Review and Research Hypothesis

Research on the risk of poverty in old age has examined both theoretical and empirical dimensions. The former focuses on the sociological theory of aging and examines the forms and effects of aging at both the individual and societal levels. Exchange theory suggests that human interaction is essentially a process of exchange and that poverty in old age represents an unequal exchange of resources [4]. Life course theory points out that old-age poverty is a challenge in the last stage of the life course, and the required resources and development needs are related to social development [5]. Borsch-Supan [6] applies the theory of social exclusion across domains to the study of aging, pointing out that the combination of the pattern of population aging, persistent economic instability, and the vulnerability of the aging population increases inequality. Through empirical research, Stephan K [7] found that the poverty rate of the elderly is associated with whether they have a partner or not. Some scholars have also focused on the measurement of the poverty aspect of the elderly, but mainly on the measurement and statistical investigation of the poverty rate of the elderly; one example is Robert Haveman [8], who took a diachronic approach to study the economic status of the worker group before and after retirement and determined that it is not only affected by age but also by gender and partner status.

By summarizing the research results of the above literature, it can be found that the main sources of poverty risk for the elderly are the economy, digital technology, and the living environment. Therefore, there may be a close relationship between regional smart development and the risk of poverty for the elderly.

Regional smart development is an essential driving force for improving the well-being of residents. Although some studies have shown that most elderly people have difficulty in using digital technology such that they can fully enjoy its benefits, both quality of life and the ability of the elderly population to socially integrate have been significantly improved by smart aging in regions with better smart development, such as present-day Beijing. Urban theory [9] points out that regions with resource and technological advantages are better able to benefit residents in the process of intellectualization. At present, China's information barriers between regions are difficult to eliminate [10], which seriously hinders the indirect spillover effect of regional smart development into rural areas, and from a macro perspective, this disparity, due to the imbalance between urban and rural development, can be extended to city to city and region to region. China's unique traditional culture means that the majority of elderly people have a stronger concept of family [11], and the degree of harmony in family relationships has a direct impact on their sense of well-being. Furthermore, social support is born in the social network [12]; the more society pays attention to the elderly group, the more perfect the social security system, and the chances of poverty in old age will be reduced accordingly. Similarly, at the micro/individual level, intergenerational support from children will significantly improve the quality of life of the elderly. Based on the above, the following research hypotheses are proposed:

**Hypothesis 1:** *Regional smart development has a significant impact on the risk of old-age poverty.*

**Hypothesis 2:** *The impact of regional smart development on the risk of old-age poverty differs somewhat across regions.*

**Hypothesis 3:** *There is heterogeneity in the impact of regional smart development on the risk of poverty in old age in terms of social support and number of children.*

### 3. Data Sources and Evaluation Indicator System

*3.1. Data Sources and Processing*

Considering the availability of data before and after the epidemic, data from China Family Panel Studies (CFPS) in 2018 and 2020 were selected for a comparative study. This dataset is an open dataset in China, aiming to collect data at the three levels of the individual, the family, and the community through tracking. It reflects social, economic, demographic, educational, and health changes in China and provides a database for academic research and public policy analysis. CFPS focuses on the economic and noneconomic well-being of Chinese residents, as well as a number of research topics including economic activity, educational outcomes, family relations and dynamics, migration, health, etc. CFPS is a national, large-scale, and multidisciplinary social tracking project. The CFPS sample covers 25 provinces/municipalities/autonomous regions, the target sample size is 16,000 households, and the survey objects include all the family members in the sample households, which can be accessed by anyone. In order to ensure the veracity of the respondents' information, we will anonymize and desensitize personally identifiable and sensitive information during the study to protect the privacy of the respondents. We will follow the guidelines of research ethics to ensure the lawful use of data and protect the rights and interests of the respondents.

In terms of data processing, the perception of poverty risk among the elderly is subject to variation based on individual characteristics, including gender, age, and education level. To better understand the relationship between regional smart development and poverty risk, the collected data were categorized into two groups: individuals aged 60–74 (considered as low old age) and individuals aged 74 and above (considered as high old age), following the age definition provided by the World Health Organization [13]. Due to the sufficient amount of data, we directly deleted the outliers and missing values in the data (such as missing indicators in the data, respondents' failure to respond, and abnormal data values, etc.), then we matched the two-year data according to the ID of the respondents, and finally, we obtained 4090 examples sample data, all of which were people aged 60 and above, covering all regions of China. Thus, our study has good representation.

Within the indicator evaluation system, due to the different dimensions and orders of magnitude of each evaluation indicator, when there is significant disparity in the indicator levels, analyzing with the original indicator values accentuates the role of indicators with higher numerical values in a comprehensive analysis. Therefore, it is necessary to standardize the original data. Considering that the BP (backpropagation) algorithm commonly uses the min–max method for data normalization, this method is also applied here for the standardization of data (Equation (1)); for indicators in the system, such as energy consumption, government burden, and others where the direction of impact is negative, a reverse standardization process is conducted (Equation (2)).

$$y_i = \frac{x_i - \min\{x_j\}}{\max\{x_j\} - \min\{x_j\}}, 1 \leq i \leq n, 1 \leq j \leq n \tag{1}$$

$$y_i' = \frac{\max\{x_j\} - x_i}{\max\{x_j\} - \min\{x_j\}}, 1 \leq i \leq n, 1 \leq j \leq n \tag{2}$$

*3.2. Evaluation Method Selection*

Due to the limited processing capability of the entropy method for high-dimensional data and its high requirements for data quality, this method falls short when dealing with certain datasets. On the other hand, artificial neural networks, as a type of nonlinear model with strong generalization and fault tolerance abilities, can effectively resolve these issues. Therefore, to determine a comprehensive value, we will combine the entropy method with artificial neural networks. This integration aims to capture and model complex indicator relationships to obtain a true and effective comprehensive value.

The backpropagation (BP) algorithm is a widely used neural network technique for tasks such as pattern recognition, function approximation, and forecasting [14]. It features good nonlinear mapping functions and high structural flexibility. The neural network can be trained according to the error backpropagation algorithm to adjust the weights, allowing the network to make accurate predictions based on the input data. The basic structure of a single hidden layer neural network is shown in Figure 1.

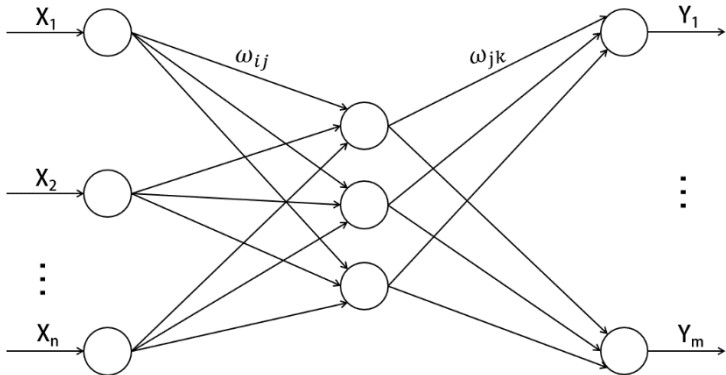

**Figure 1.** BP neural net topology.

The forward propagation process is mainly for the input layer to input signals, the hidden layer to process information, and finally, the output layer to output values. For the ith neuron node in the hidden layer [15], the following apply:
Input signal:

$$net_j = \sum_{i=1}^{N} \omega_{ij}\mu_i - \theta_j; \tag{3}$$

Output signal:

$$o_j = f(net_j) = f(\sum_{i=1}^{N} \omega_{ij}\mu_i - \theta_j). \tag{4}$$

For the kth neuron node of the output layer, the following apply:
Input signal:

$$net_k = \sum_{j=1}^{L} (\omega_{jk}o_j - \theta_k); \tag{5}$$

Output signal:

$$y_k = f(net_k) = f(\sum_{j=1}^{L} (\omega_{jk}o_j - \theta_k)), \tag{6}$$

where $\omega_{ij}$ represents the weight of the neural nodes connecting the input layer and the hidden layer, $\omega_{jk}$ represents the weight of the neural nodes of the hidden layer and the output layer, and $\theta_k$ is the threshold. f (-) is the activation function, which is here set as the sigmoid function.

$$f(x) = \frac{1}{1+e^{-x}} \tag{7}$$

The backpropagation algorithm involves propagating error signals from the output layer to the input layer, adjusting the weights and biases of each layer along the way. Specifically, this process entails calculating the error signal at the output layer, transmitting it backwards to the hidden layers, computing the error signals for each neuron in these layers, and updating the weights and biases based on these error signals using the gradient descent method. These steps are repeated until convergence or a predetermined number of training iterations is reached.

Suppose there are a total of P training samples, and $y_k^{(P)}$ and $d_k^{(P)}$ are the actual output and expected output of the kth neuron node of the output layer, respectively, then the error signal generated by the output of neuron j is defined as

$$e = d_k^{(P)} - y_k^{(P)}. \tag{8}$$

Minimizing the root-mean-square difference makes the function continuously differentiable; then, the instantaneous error energy of neuron j is

$$E_k^{(P)} = \frac{1}{2}e^2. \tag{9}$$

The total instantaneous error energy of the entire network is

$$E = \frac{1}{2}\sum_{p=1}^{P}\sum_{k=1}^{M}e^2. \tag{10}$$

According to the gradient descent algorithm, the connection weights and thresholds are adjusted layer by layer. The connection weights from the hidden layer to the output layer are modified to $\Delta W$, and the adjustment equation is as follows:

$$\Delta\omega_{jk} = -\eta\frac{\partial E}{\partial\omega_{jk}} = -\eta\frac{\partial E}{\partial e}\frac{\partial e}{\partial y_k}\frac{\partial y_k}{\partial net_k}\frac{\partial net_k}{\partial\omega_{jk}}. \tag{11}$$

The partial derivative $\frac{\partial E}{\partial\omega_{jk}}$ represents a sensitivity factor that determines the search direction of synaptic weight $\omega_{jk}$ in the weight space. Take the differential of e on both sides of Equation (7) to obtain $\frac{\partial E}{\partial\omega_{jk}} = e$; take the differential of $y_k$ on both sides of Equation (6) to obtain $\frac{\partial e}{\partial y_k} = -1$; take the differential of $net_k$ on both sides of Equation (4) to obtain $\frac{\partial y_k}{\partial net_k} = f'(net_k)$; and finally, take the differential of $\omega_{jk}$ on both sides of Equation (3) to obtain $\frac{\partial net_k}{\partial\omega_{jk}} = o_j$. Therefore, the final value of the correction can be obtained as

$$\Delta\omega_{jk} = \eta\delta_k o_j, \tag{12}$$

where $\eta$ represents the learning rate, and $\delta_k$ is the local gradient obtained according to the delta rule, indicating the required change in synaptic weight. The specific equation is

$$\delta_k = \sum_{p=1}^{P}\sum_{k=1}^{M}(d_k^{(p)} - y_k^{(p)})f'(net_k). \tag{13}$$

Similarly, the threshold correction value of the output layer $\Delta\theta_k$, the connection weight correction value of the input layer to the hidden layer $\Delta\omega_{ij}$, and the threshold correction value of the formula hidden layer $\Delta\theta_j$ are similar to those in Equation (9). After differentiating the corresponding values, we obtain the following:

$$\Delta\theta_k = -\eta\frac{\partial E}{\partial\theta_k} = -\eta\frac{\partial E}{\partial y_k}\frac{\partial y_k}{\partial net_k}\frac{\partial net_k}{\partial\theta_k} = \frac{\eta}{k}\delta_k, \tag{14}$$

$$\Delta\omega_{ij} = -\eta\frac{\partial E}{\partial\omega_{ij}} = -\eta\frac{\partial E}{\partial o_j}\frac{\partial o_j}{\partial net_j}\frac{\partial net_j}{\partial\omega_{ij}} = \eta\delta_j\mu_i, \tag{15}$$

$$\Delta\theta_j = -\eta\frac{\partial E}{\partial\theta_j} = -\eta\frac{\partial E}{\partial o_j}\frac{\partial o_j}{\partial net_j}\frac{\partial net_j}{\partial\theta_j} = \eta\delta_j. \tag{16}$$

Among them:

$$\delta_j = \sum_{p=1}^{P} \sum_{k=1}^{M} \omega_{jk}(d_k^{(p)} - y_k^{(p)}) f'(net_k) f(net_j). \tag{17}$$

The above process is a standard BP network. In order to further improve the performance of standard BP, momentum factor $\alpha$ is introduced, so the weighting equation of BP network is modified as follows:

$$\Delta\omega_{jk} = \partial\Delta\omega_{jk}(n-1) + \eta\delta_k o_j. \tag{18}$$

*3.3. Evaluation Index System Construction and Result Analysis*

Literature research shows that the highest proportion of economically poor older people face multidimensional poverty risks at the same time [16]. There is a strong correlation between an individual's economic strength and their poverty risk. In addition, family endowment and social capital (family endowment includes economic resources, education level, occupational status, etc., while social capital refers to social network, social support, and social participation) have a moderating effect on the dimensions of poverty risk in old age; for example, a higher family endowment may increase the chances of older people escaping poverty, and a good social network and social support can provide older people with more resources to help them cope with life difficulties and poverty risks [17]. Therefore, combining the results of the existing literature, we constructed an index system of five dimensions, namely, economic risk, psychological risk, family risk, individual risk, and social risk, and assigned weights to the corresponding indexes via the entropy value method and performed normalization. Finally, we obtained the evaluation index system of the level of poverty risk in old age, which is shown in Table 1. After substituting the corresponding data according to the calculated weights, the corresponding index scores of poverty-causing risks are obtained, and the poverty-causing risk is classified. The text is divided into five grades based on the grading standard of Ebbinghaus [18], and values are assigned to each grade accordingly.

**Table 1.** System of indicators for evaluating the risk of poverty among the elderly population.

| Layer | Index | Direction | Weight Coefficient |
| --- | --- | --- | --- |
|  | All kinds of subsidies to after-tax monthly income | (-) | 0.120 |
| Economic risk | Difference in household transfers | (-) | 0.105 |
|  | Total medical expenditure | (+) | 0.105 |
| Psychological risk | Degree of loneliness | (+) | 0.094 |
|  | Degree of happiness | (-) | 0.094 |
| Individual risk | Degree of health | (-) | 0.091 |
|  | Level of intelligence | (-) | 0.063 |
| Family risk | Have a family to live with | (-) | 0.049 |
|  | Level of family care input | (-) | 0.014 |
|  | Social welfare | (-) | 0.105 |
| Social risks | Awareness of social issues | (+) | 0.082 |
|  | Social connections | (-) | 0.079 |

After the above weights are determined from Table 1, the comprehensive evaluation index Y is obtained by using Equation (3), and it is used as the output in the subsequent evaluation model.

$$Y = \sum_{i=1}^{12} X_i W_i \tag{19}$$

The comprehensive evaluation model is obtained through BP model, and the specific equation is as follows:

$$Y = f\left\{ \left[ f(\omega_{ij} \cdot \mu_i - \theta_j) \right] \cdot \omega_{jk} - \theta_k \right\}, \tag{20}$$

It is very important to select a suitable parameter configuration for BP algorithm training and model performance. After debugging the model and referring to the literature, the maximum epoch is set to 1000. The minimum error of the training target is set to 0.000001. The learning rate is set to 0.1. In addition, in order to solve the problem that the error caused by the gradient descent method is not sensitive to weight, momentum factor is introduced to optimize BP, and the value is set to 0.9. Because the number of hidden layer nodes determines the complexity and expressiveness of the neural network model, too many hidden layer nodes may lead to overfitting of the model, and too few hidden layer nodes may lead to underfitting of the model. The selection of the appropriate number of nodes in the hidden layer needs to consider the complexity of data and the fitting ability of the model. Here, the empirical equation hiddennum = sqrt (m + n) + a is adopted [19], where m is the number of nodes in the input layer, n is the number of nodes in the output layer, and a is generally an integer between 1 and 10, which is selected through the 50% cross-validation method. The optimal number of hidden layer nodes is nine. Under this parameter setting, the training error and test error are shown in Figure 2.

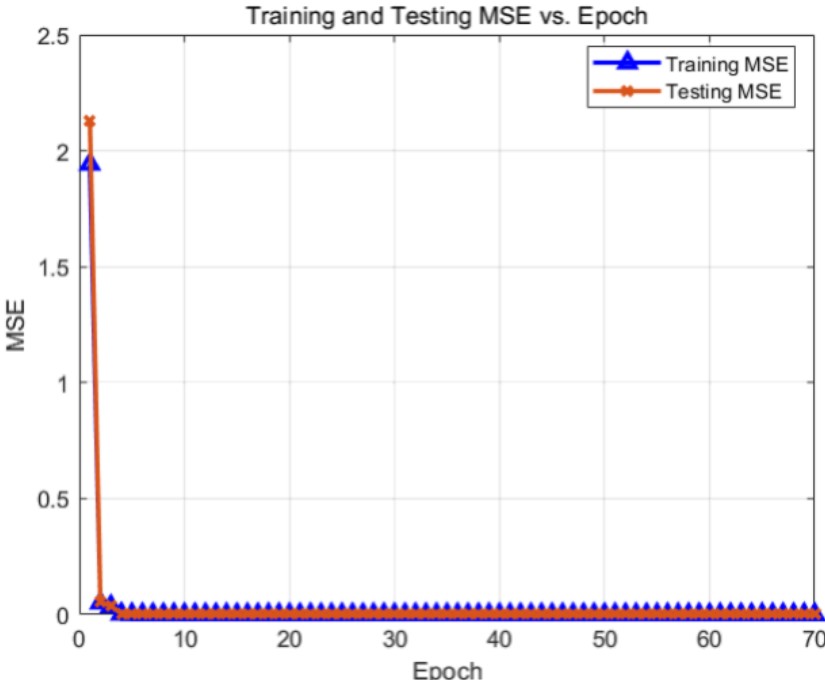

**Figure 2.** Iterative error.

The respondents' regions were divided into three parts according to geographical location—eastern region, central region, and western region—and the risk of elderly people becoming poor was calculated, respectively [20]. The comparison results are shown in Figure 3. In the overall dimension of 2020, the risk of poverty among China's elderly population presents a distribution of "high in the middle and low on both sides", and more than half of the elderly respondents' risk of poverty is a general risk, which indicates that the risk of poverty among most elderly people is still in a controllable stage. From a regional perspective, the risk of poverty among the elderly in the eastern region is lower than that in the central and western regions, and it is preliminarily believed that the risk of poverty among the elderly is related to the development level of local economy, science, and technology.

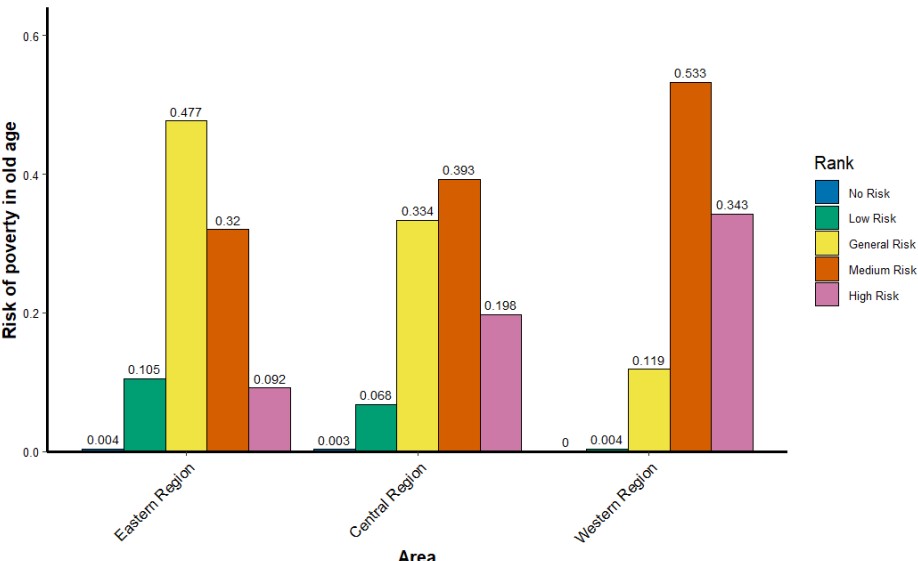

**Figure 3.** Regional comparison of elderly poverty risk in China, 2020.

*3.4. Regional Smart Level Evaluation Index System*

Current measures related to regional smart development are mainly designed around smart cities. Giffinger R [21] suggests that smart cities need to be measured with economy, mobility, environment, population, and government as key dimensions, with full consideration of the importance of ecology, economy, and manpower to the process of urban smart city development. J. M. Eger [22] states that the keys to smart city development are technological progress, economic development, job growth, and improved quality of life for residents, and holds that smart technology and residents' life should be included in the index system. Hongbo S [23] established a comprehensive evaluation index system for the smart development of cities with a PSF evaluation model based on urban system theory, but smart humanities, scientific and technological innovation capability, and green development capability were not included in the index system.

Based on the above literature, and considering the limitations and deficiencies of the index system currently constructed, the core essence of "people-oriented" smart cities is included in the measurement system as an indicator. Additionally, negative indicators such as crime rate and unemployment rate are incorporated to provide a comprehensive assessment of the overall smartness of a region. Consequently, a regional smart evaluation indicator system is developed, comprising six first-level indicators, sixteen s-level indicators, and thirty-four third-level indicators. This system is constructed based on three dimensions: the foundation layer, the application layer, and the target layer. For further details, please refer to Table 2.

**Table 2.** Regional smart level evaluation index system.

| Layer | Primary Index | Secondary Index |
| --- | --- | --- |
| Smart technology | Foundation of technology | (X1) Number of Internet users per 100 people |
| | | (X2) The overall level of Internet development |
| | Ability to innovate | (X3) Technology market turnover |
| | | (X4) Digital economy patent application data |
| Smart ecology | Environmental condition | (X5) Technology market turnover |
| | | (X6) Digital economy patent application data |
| | | (X7) Harmless treatment rate of household garbage |
| | Ecological response | (X8) Daily treatment capacity of urban sewage |
| | | (X9) Area of afforestation in the current year |

**Table 2.** *Cont.*

| Layer | Primary Index | Secondary Index |
|---|---|---|
| Smart human capital | Education and cultivation | (X10) The intensity of R&D investment |
| | | (X11) The proportion of education expenditure in the general budget of local finance |
| | | (X12) Persons employed in urban units in scientific research and technology services |
| | Human capital | (X13) Persons employed in urban units in information transmission, software, and information technology services |
| Smart economy | Vitality of economic development | (X14) Digital inclusive financial index |
| | | (X15) Gross product |
| | | (X16) Revenue from information technology services |
| | Green development | (X17) Energy consumption per unit GDP |
| | | (X18) Green finance index |
| | Income level of residents | (X19) Per capita disposable income of permanent urban residents |
| Smart governance | Urban governance | (X20) Proportion of information security revenue |
| | | (X21) Local fiscal expenditure on public security |
| | Government service | (X22) Investment in environmental pollution control in GDP |
| | | (X23) Coverage of social service institutions |
| | Medical conditions | (X24) Number of beds in medical and health institutions |
| | | (X25) Consumer price index of medical services per 10,000 population |
| Smart people's livelihood | Construction of culture | (X26) The number of public library collections per capita |
| | | (X27) The number of public library institutions |
| | | (X28) The level of urban infrastructure |
| | Infrastructure construction | (X29) The number of bus-electric vehicles in operation |
| | | (X30) Highway density |
| | | (X31) Income from basic endowment insurance fund |
| | Social security | (X32) Income from urban basic medical insurance fund |
| | | (X33) Number of people participating in unemployment insurance |
| | People's happiness | (X34) Provincial crime rate |
| | | (X35) Registered urban unemployment rate |

The regional smartening indicators also use the entropy value method to assign weights to the corresponding indicators and perform normalization to obtain the development level of smartening in China, which is shown in Table 3. To present the spatial and temporal evolution trend of the development level of the indicators more intuitively, only the three major regions of the east, central, and western regions and the national scope of the relevant trends are listed. As can be seen in Table 3, during the period of 2018–2020, the national level of smart development shows a growth trend, but the growth rate is not fast. From the regional perspective, the development level of each region in 2018 is as follows: the eastern region, 0.525; the central region, 0.339; and the western region, 0.302. However, in 2020, the index will show a different order: the eastern region, 0.560; the central region, 0.384; and the western region, 0.345. It can be seen that the region with better economic development has a higher level of exponential development due to the accumulation of high-tech talents, strong innovation ability, and rich development resources, while the western region, as a relatively underdeveloped region, has a lower level of development. This is due to the relatively weak infrastructure construction, industrial structure, and innovation capacity in the western region, as well as the lack of high-tech talents and innovation resources. At the same time, the geographical location and traffic conditions of the western region may also limit the speed and scope of its economic development.

In order to preliminarily understand the relationship between regional smart development and elderly poverty risk, we took the calculated risk value of elderly poverty risk in each province of China and conducted a correlation analysis of the two. The results showed that there was a certain correlation between the two (Figure 4).

**Table 3.** Regional level of smart development.

| Year | Region | Intelligent Level | Smart Technology | Smart Ecology | Smart Human Capital | Smart Economy | Smart Governance | Smart People's Livelihood |
|---|---|---|---|---|---|---|---|---|
| 2018 | Nationwide | 0.393 | 0.361 | 0.660 | 0.224 | 0.314 | 0.375 | 0.418 |
| | Eastern region | 0.525 | 0.555 | 0.760 | 0.356 | 0.467 | 0.459 | 0.532 |
| | Central region | 0.339 | 0.285 | 0.646 | 0.163 | 0.256 | 0.264 | 0.418 |
| | Western region | 0.302 | 0.222 | 0.571 | 0.136 | 0.204 | 0.373 | 0.303 |
| 2020 | Nationwide | 0.434 | 0.462 | 0.635 | 0.295 | 0.348 | 0.440 | 0.425 |
| | Eastern region | 0.560 | 0.657 | 0.753 | 0.462 | 0.517 | 0.454 | 0.518 |
| | Central region | 0.384 | 0.379 | 0.602 | 0.210 | 0.282 | 0.383 | 0.450 |
| | Western region | 0.345 | 0.327 | 0.541 | 0.190 | 0.228 | 0.468 | 0.314 |

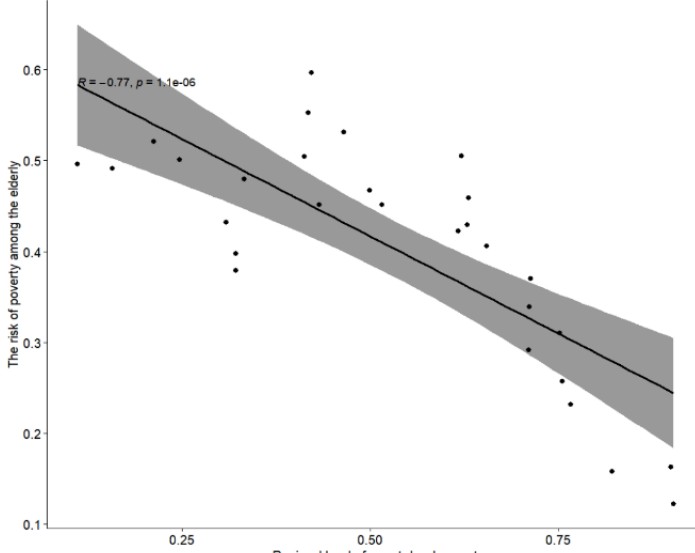

**Figure 4.** Correlation test.

## 4. Effect of Regional Smart Development on the Risk of Poverty in Old Age

### 4.1. Model Construction

A fixed effects model was chosen with the following expression [24]:

$$EPR_{it} = C + \alpha_1 RIL_{it} + \gamma_1 X_{it} + \mu_i + \lambda_t + \varepsilon_{it}, \qquad (21)$$

where the explanatory variable $EPR_{it}$ denotes the level of old-age poverty risk in year t for the ith respondent, $RIL_{it}$ denotes the level of intellectualization of the area where the ith respondent lives in year t, $X_{it}$ is the control variable, $\mu_i$ denotes individual fixed effects, $\lambda_t$ denotes time fixed effects, and $\varepsilon_{it}$ is a randomized disturbance term. The setting of the control variables mainly examined the individual level of the respondents, including gender, education level, age, and whether they had pension insurance, etc., and the household level mainly considered the intergenerational relationship of the respondents [25].

### 4.2. Regression Analysis

Following the fixed effects model, regression estimation was conducted to examine the relationship between the level of regional smart development and the risk of poverty in old age. The specific regression results are shown in Table 4. Considering that the collected data may have heteroscedasticity problems [20], the clustering robust standard error corrected based on individual ID coding is used according to the method of Chengfeng Yu [26]. In

Model (1), no control variables are added, and on the assumption that the individual fixed effects are controlled, the impact of regional smart development level on the risk of poverty in old age passes the 1% statistical test, and the regression result is significantly negative, with the regression coefficient being −2.153. A number of control variables are added to Model (2), and individual fixed effects are controlled. The results show that the impact of regional smart development level on the risk of elderly poverty is still significantly positive at the level of 1%. Therefore, Model (2) is taken as the benchmark regression, and it can be seen that the addition of control variables or the improvement of regional smart development level will effectively reduce the risk of elderly poverty. This result can be considered robust.

**Table 4.** Specific regression results.

| Variable | Model (1) The Elderly | Model (2) The Elderly | Model (3) The Elderly in the High Age Group | Model (4) The Elderly in the Low Age Group | Model (5) Eastern Region | Model (6) Central Region | Model (7) Western Region |
|---|---|---|---|---|---|---|---|
| Intelligent level | −2.153 *** | −0.244 *** | −0.170 ** | −1.018 ** | −0.345 *** | −0.743 *** | 0.117 ** |
| Urban and rural areas | | −0.013 * | −0.009 * | 0.007 | 0.001 | 0.014 | 0.013 |
| Sex | | 0.020 ** | 0.014 ** | 0.037 * | 0.009 * | −0.005 | 0.005 * |
| Level of education | | −0.016 *** | −0.017 *** | −0.026 ** | −0.008 * | 0.012 ** | −0.018 ** |
| Age | | −0.048 *** | −0.048 *** | −0.046 *** | −0.057 *** | −0.033 *** | −0.032 *** |
| Retirement Security | | −0.040 *** | −0.004 * | −0.029 ** | −0.003 | 0.014 | −0.011 |
| Marital status | | 0.018 | 0.039 | −0.032 | 0.023 | 0.016 | 0.010 |
| Intergenerational relations | | −0.005 ** | −0.005 * | −0.022 *** | −0.007 * | −0.005 | −0.001 |
| Constant | 1.611 *** | 3.998 *** | 3.920 *** | 4.787 *** | 4.274 *** | 3.190 *** | 3.356 *** |
| Individual fixed effects | Controlled | Controlled | Controlled | Controlled | Controlled | Controlled | Controlled |
| Value of observation | 4090 | 4090 | 3531 | 559 | 1830 | 1268 | 992 |
| R2 | 0.262 | 0.366 | 0.367 | 0.421 | 0.329 | 0.389 | 0.453 |

Note: * denotes $p < 0.1$; ** denotes $p < 0.05$; *** denotes $p < 0.01$.

Further divided by the age of the respondents, the regression results of the elderly in the high age group and the elderly in the low age group are obtained. The results show that the level of regional smart development has a significant impact on both groups, and the impact on the poverty risk of the elderly is more obvious, with the regression coefficient being −1.108, which is also consistent with the previous analysis; that is, with the rapid development of China's economy and the continuous improvement of the level of urban intelligent construction, the quality of life and psychological status of the elderly have been well guaranteed. In addition, from the results of the control variables, age and intergenerational relationship levels significantly affect the risk of poverty among the elderly.

*4.3. Endogeneity Discussion*

Regarding the potential endogeneity of the variables, the selection method proposed by Shariatpour [27] is used to address this issue. In this approach, the level of innovation and entrepreneurship in each region is used as an instrumental variable (IV) to capture the level of regional smart development [28]. The level of innovation and entrepreneurship serves as an indicator of the region's development in terms of scientific and technological innovation, entrepreneurial environment, talent attraction, and supportive policies.

Table 5 shows the regression results based on the two-stage least-squares method, where column 1 is the variable name, column 2 is the first-stage regression result, and column 3 is the second-stage regression result, and column 4 is the regression result of the fixed effect model. In the first-stage regression, the regression coefficient of IV and RIL was 0.115 and significantly positive at the 1% level; that is, IV met the correlation conditions of

instrumental variables and could fully explain the core variables, and since the value of Cragg–Donald Wald F-statistic is 1070.55, there is no weak instrumental variable problem. In the second-stage regression, after using IV to control for endogeneity, the level of regional smart development can still significantly reduce the risk of poverty among the elderly, and the regression coefficient value of both is −0.153, which is significantly negative at the 1% level. Comparing the results of the second-stage regression with those of the fixed-effects model in the baseline regression, the absolute value of the coefficient for the core explanatory variable decreased from 0.244 to 0.153. This suggests that the baseline regression model underestimated the treatment effect, indicating the need for control. It is important to note that the degree of underestimation did not exceed 0.1, indicating that the endogeneity problem was not severe.

**Table 5.** IV regression results.

| | (1) Level of Intelligent Development | (2) Risk of Poverty among Older Persons | (3) Fixed Effect Model |
|---|---|---|---|
| IV | 0.115 *** | | |
| RIL | | −0.153 *** | −0.244 *** |
| Urban and rural areas | 0.009 *** | −0.007 | −0.013 * |
| Sex | 0.003 | 0.016 *** | 0.020 ** |
| Level of education | −0.003 | −0.015 *** | −0.016 *** |
| Age | 0.000 | 0.000 | −0.048 *** |
| Retirement Security | −0.005 | −0.034 *** | −0.040 *** |
| Marital status | −0.007 * | 0.017 ** | 0.018 |
| Intergenerational relations | −0.001 | −0.020 *** | −0.005 ** |
| Constant | −0.938 *** | 0.680 *** | 3.998 *** |
| Wald test | | 204.64 | |
| F | 1070.55 | | |
| P | | 0.0000 | |
| Individual fixed effects | Controlled | Controlled | Controlled |
| Value of observation | 4090 | 4090 | 4090 |
| R2 | 0.690 | 0.245 | 0.366 |

Note: * denotes $p < 0.1$; ** denotes $p < 0.05$; *** denotes $p < 0.01$.

### 4.4. Robustness Test

The robustness test is conducted by analyzing the above regression. In the first instance, regional smart development replaces the explanatory variables, and the sub-indicators of the level, namely, the comprehensive development level of the Internet and basic network construction, are adopted to replace the core explanatory variables. In the second instance, in replacing the explanatory variable, economic poverty is the most concerning issue pertaining to elderly poverty, so the family economic situation is used as the replacement variable of the explanatory variable. The robustness test is shown in Table 6, where Model (1) and Model (2), respectively, use the comprehensive development level of the Internet and basic network construction as explanatory variables, while Model (3) takes the family economic situation as the explanatory variable. The regression results show that there is a positive correlation between the level of regional intelligent development and the family economic status of the elderly. This correlation has a significant promoting effect, indicating that as the level of smart development advances, the economic status of elderly households improves. In addition, the improvement of Internet development and the expansion of basic network infrastructure can significantly reduce the risk of elderly poverty. The regression results show the robustness of this relationship.

**Table 6.** Robustness test results.

| Variable | Risk of Poverty among Older Persons | | Family Financial Situation |
|---|---|---|---|
| | Model (1) | Model (2) | Model (3) |
| RIL | | | 0.011 ** |
| The comprehensive development level of the Internet | −0.442 ** | | |
| Basic network construction | | −0.008 *** | |
| Variable of control | YES | YES | YES |
| Individual fixed effects | Controlled | Controlled | Controlled |
| Value of observation | 4090 | 4090 | 4090 |
| R2 | 0.366 | 0.372 | 0.762 |

Note: * denotes $p < 0.1$; ** denotes $p < 0.05$; *** denotes $p < 0.01$.

### 4.5. Heterogeneity Test

Regional heterogeneity was further examined based on the eastern, central, and western regions [29], and the findings are presented in Table 7. The eastern and central regions exhibit a higher level of smart development, which significantly reduces the risk of poverty among the elderly at a significance level of 1%. On the one hand, this is because their economic development and resource allocation are more advanced; and on the other hand, it may be because regional smart development has optimized the local socioeconomic structure and living environment, thereby reducing the risk of poverty for the elderly. Conversely, in the western region, an increase in smart development levels significantly raises the risk of poverty among the elderly. This is primarily due to two reasons: first, inadequate development hinders digital literacy among older adults and prohibits them from keeping pace with digital advancements and information-driven lifestyles, leading to social and psychological impoverishment; second, since emerging industries have not yet fully formed in this region, a complex economic environment ensures, with limited employment opportunities for seniors who may face lifestyle changes or health issues, thereby increasing their vulnerability to poverty.

**Table 7.** Results of regional heterogeneity test.

| Variable | (1) Eastern Region | (2) Central Region | (3) Western Region |
|---|---|---|---|
| RIL | −0.345 *** | −0.743 *** | 0.117 ** |
| Variable of control | YES | YES | YES |
| Individual fixed effects | Controlled | Controlled | Controlled |
| Constant | 4.274 *** | 3.190 *** | 3.356 *** |
| Value of observation | 1830 | 1268 | 992 |
| R2 | 0.329 | 0.389 | 0.453 |

Note: * denotes $p < 0.1$; ** denotes $p < 0.05$; *** denotes $p < 0.01$.

In addition, in order to test the heterogeneity of the impact of smart development on the poverty risk of the elderly under different social support structures [30], this paper discusses it from two levels: informal support and formal support (Table 8). The dimension of formal support is measured by participation in social insurance. In the dimension of informal support, if the elderly receive financial support or care from their children, improvement in the regional smart development level can significantly reduce the risk of poverty among the elderly, and the impact of economic support is greater. In terms of formal support, the elderly who participate in social insurance are less likely to fall into poverty in the era of rapid smart development, and effective social support can significantly reduce the risk of poverty among the elderly. The above regression results confirm our research hypothesis.

**Table 8.** Social support heterogeneity test.

| Variable | (1) Receive Financial Support | (2) No Financial Support Is Received | (3) Receive Care Support | (4) No Care Support | (5) Have Social Insurance | (6) No Social Insurance |
|---|---|---|---|---|---|---|
| RIL | −0.496 ** | 0.585 ** | −0.293 ** | 0.413 * | −0.234 ** | −0.218 |
| Variable of control | YES | YES | YES | YES | YES | YES |
| Individual fixed effects | Controlled | Controlled | Controlled | Controlled | Controlled | Controlled |
| Constant | 3.712 *** | 3.897 *** | 3.899 *** | 3.732 *** | 3.857 *** | 5.461 *** |
| Value of observation | 2325 | 1765 | 1620 | 2470 | 3538 | 552 |
| R2 | 0.360 | 0.345 | 0.348 | 0.367 | 0.350 | 0.462 |

Note: * denotes $p < 0.1$; ** denotes $p < 0.05$; *** denotes $p < 0.01$.

## 5. Conclusions and Suggestions

### 5.1. Research Conclusions

Based on the above analysis, the following conclusions are drawn:

First, China's current level of smart development is experiencing continuous growth; however, there is still potential for improvement in the growth rate. In addition, the regional level of smartness shows a gradual upward trend from east to west. The economically developed eastern and central regions are currently enjoying a higher level of development, while the overall level in the western region is rising noticeably. The overall poverty risk of the elderly population shows a distribution pattern characterized by "high in the middle and low at both ends". More than half of the elderly respondents face a general risk of poverty, while a small minority of elderly people are not at risk of poverty. In addition, the western region has a higher proportion of medium- and high-risk groups than other regions.

Second, improving the regional level of smart development can effectively reduce the vulnerability of the elderly to poverty, with different levels of influence observed in different regions. In regions with advanced smart development, increasing the smartness level can effectively reduce the risk of elderly poverty. Conversely, in regions with inadequate smart development, the risk of poverty will escalate as the smartness level improves. Specific to the current situation of smart city development in various provinces in China, the smart development in the eastern and central regions is in the stage of improving the quality of life of the elderly, while the smart development in the western regions is in the stage of reducing the quality of life of the elderly. The inadequate development of regional smart development means that the elderly face higher poverty risk. On the other hand, the impact of regional smart development on the risk of poverty in old age varies depending on the level of social support. If the elderly receive financial support or care from their children, the risk is significantly reduced as the level of intelligence increases. Elderly people's participation in social insurance programs increases their ability to adapt to a progressively smart lifestyle.

### 5.2. Relevant Suggestion

Based on the above research conclusions, the following suggestions are put forward:

First, it is imperative to actively promote the growth of regional economies and effectively implement smart development strategies tailored to each region. Considering the current situation of smart development in all regions of China, promoting regional economies is still the top priority at present. At the same time, each region should correctly assess the different impacts of smart development on the poverty risk of the elderly and formulate appropriate improvement policies for different regions based on the current smart development situation and the priority needs of the elderly. For example, in regions

with a high level of smart development, priority can be given to the development of smart industries that involve the elderly and provide employment opportunities. In regions with low cognitive ability, it is imperative to prioritize the development of intelligent infrastructure and the widespread use of information technology. This approach will effectively enhance the elderly population's ability to access and benefit from intelligent conveniences.

Second, it is necessary to accelerate the process of digital construction to improve the user experience and convenience of digital facilities. Digital construction plays a crucial role in promoting intelligent regional development as it has the potential to significantly improve the digital literacy and social engagement of the elderly population. All regions should accelerate the digitization of conventional infrastructure, taking into account the needs of the elderly, and prioritize the incorporation of human-centered design principles into the construction process. These can include the implementation of features such as an elderly-mode key switch, logistics home services, and the use of elderly cards, among others. In addition, as part of the digital transformation process, it is imperative to enhance support for the elderly population and promote their ability to use smart technologies. This will enable them to adapt more easily to, and participate in, a smart society.

Third, it is imperative to strengthen the social security mechanism for the elderly population and provide them with adequate social attention and support. Given the importance of social support, in order to effectively reduce the risk of poverty among the elderly, formal and informal support should be developed at two levels. On the one hand, it is necessary to speed up the establishment of a comprehensive social security system for the elderly, expand the scope and variety of social insurance for the elderly, and promote the development of technologically advanced elderly care services in communities by establishing elderly dining halls and creating an age-friendly environment; on the other hand, we should focus on and support the elderly and empty-nesters, provide elderly care services and elderly care subsidies, promote child care and economic support, and ensure that the needs of the elderly in the information environment are met.

### 5.3. Possible in-Depth Research in the Future

Considering the negative effects caused by inadequate urban smart development, there may be a nonlinear relationship between smart development and the risk of poverty in the elderly, which is most likely an inverted U-shaped relationship. Future studies can be conducted from this perspective to explore the critical point of development of the two variables.

**Supplementary Materials:** The following supporting information can be downloaded at: https://www.mdpi.com/article/10.3390/su16073094/s1.

**Author Contributions:** In this study, C.L. improved the research method and logic of the paper; H.L. prepared research data for this study and analyzed the data; and L.S. conceived and designed the study framework and provided the method. The manuscript was produced through contributions by all authors. All authors have read and agreed to the published version of the manuscript.

**Funding:** This research received no external funding.

**Institutional Review Board Statement:** The study was conducted according to the guidelines of the Declaration of Helsinki, Ethical review and approval were waived for this study, due to this study uses China's open data set CFPS, which is accessible to all people and organizations, and the research content mainly focuses on the impact of smart development on the lives of the elderly, it does not require the approval of an ethics committee or institutional review board.

**Informed Consent Statement:** Patient consent was waived due to REASON. (CFPS is an open dataset that anyone can use.)

**Data Availability Statement:** Data are contained within the article and supplementary materials.

**Conflicts of Interest:** The authors declare no conflict of interest.

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
