# Peer review of "Impact Analysis of Regional Smart Development on the Risk of Poverty among the Elderly"

_sustainability, doi:10.3390/su16073094_

Round 1
Reviewer 1 Report
Comments and Suggestions for Authors
Well thought out methodology with great data analysis. Would like to see another paragraph on conclusions providing more insights on regionalization impact/differences along with future research.
Some minor English revisions required especially around the usage of smartization (not an English word) and regional smartness.
Comments on the Quality of English LanguageSome minor English revisions required especially around the usage of smartization (not an English word) and regional smartness. There are also some Chinese characters that need to be replaced.
Author Response
Thank you very much for your insightful and valuable comments and constructive suggestions, which have significantly improved the clarity of this paper. We have revised our paper based on your kind advice. All revisions are in a red color in the revised paper.Please refer to the attachment for details.
Brief response:
- According to your suggestions, we have added views on future feasibility studies in the conclusion (lines 500-505) and expanded the content of regional differences (lines 450-458).
- Correct the wrong words in the article
Thanks again for your advice

Reviewer 2 Report
Comments and Suggestions for Authors
the document focuses on the impact of technological development in urban infrastructures on the life quality of the elderly and on the risk of poverty faced by this social category. This study is very importante but some points need to be revised:
1. The abstract is generally well-written and contains the elements needed for the summary of the work.
2. A few spelling mistakes need to be checked.
3. Before conducting such a study, we need to see if there is a correlation between the poverty index among the elderly and the city's technological development.
4. The first hypothesis is unclear and needs to be reformulated.
5. Before deleting outliers and missing values in the database, have you studied the impact of these variables on the results (beware of deleting a parameter that has a direct impact on the target variable).
6. I think you need to introduce people's educational or intellectual level as an indicator.
7. In figure 3 you've divided people into three groups according to geographical location, but you haven't shown the impact of these regions on the risk of poverty (if these regions are different in terms of lifestyle, infrastructure, etc., this must be mentioned, otherwise I don't see the purpose of this subdivision), and in the first part you've mentioned that you're going to divide people into two groups according to age range without justifying the choice of this division! this creates confusion!
8. Please check the Chinese text in table 8.
In general, the study is well done, the results are encouraging, but it lacks a little clarification of certain points such as the choice of groups, the methodology and the suggestions section is a little vague, specific recommendations should be given for the results obtained supported by strong arguments.
Comments on the Quality of English LanguageThe quality of english is fine.
Author Response
We are very grateful to thank your constructive suggestions, which have significantly improved the clarity of this paper. We have revised our paper based on your kind advice. All revisions are in a red color in the revised paper.Please refer to the attachment for details.
Brief response:
- We have corrected the misspellings in the article and slightly modified the summary
- Correlation analysis is necessary! According to your suggestion, we have added the content of correlation analysis (line 315).
- We have redescribed the first hypothesis
- Before removing outliers and missing values, we did study the impact of these variables on the results. These variables are further analyzed to ensure that parameters that have a direct impact on the target variable are not removed.
- Intelligence level has been introduced into the individual risk dimension of the index system
- We have added the description of regional differences (lines 402-416).
- With regard to age classification, we refer to the World Health Organization's method of defining age
The remaining changes have been marked in red, thanks again for your suggestions

Reviewer 3 Report
Comments and Suggestions for Authors
Sustainability-2407730
This is an analysis of regional “smart” development in China on the risk of poverty among two groups of older adults. Using existing survey data from a national survey conducted during two time periods they developed and tested several statistical models of poverty development. This study adds to our understanding with a specific application of smart development to the problem of poverty in older age across countries.
Major concern about protection of human subjects. Not addressed in the paper nor at the end. MUST be addressed. Please see the https://www.mdpi.com/journal/sustainability/instructions#ethics
Comments:
1. Introduction and Literature Review: The authors argue well for the risk of poverty in the growing elderly population in China and make a point that smart regional development can enhance the ability of older adults to live well. The hypotheses flow from the literature reviewed. One suggestion is that it may help to use italics for the unique concepts used in smart development such as “smartinization,” “wisdom” etc. to draw the English readers understanding away from the more common meaning of such terms.
2. Data Sources and evaluation indicator system. This section needs expansion regarding the original data sets. For example, what is the total possible sample and how representative is it? What were the inclusion and exclusion criteria for the data abstracted? Are the tools they used valid and reliable? For example, did you start with 10,000 participants who met inclusion criteria but were then excluded due to missing data or being an outlier? It is not clear how you reached your final 4090 nor did you speak to the representativeness of these participants to the rest of the elderly in the data set or China. How many variables per participant did you start with? Other than the statistical methods presented what was your theory, logic or reasoning to reduce this data set to a more manageable number?
I have some concerns about the amount of data scrubbing that occurred in this study since we are not told what specific variables from the data set were used. Remember that the more you manipulate the data prior to the analysis the greater the risk of a Type 2 error and the more difficult it becomes to really understand what happened.
How did you create the regions? Geographically by what? Some other system such as a variable in the data set? Was the “ID” assigned the data source?
3. Results: Clear
4. Conclusion: Although the results are based on some very sophisticated analyses with statistical significance, they are not convincing when you consider that most of the variability could be accounted for by income and social support rather than “smartness.”
5. Tables: Suggest you left align the text portions of the tables and align the items in the columns for easier reading. As presented they are difficult to determine which index goes with which layer in Table 1, and in Table 2 same plus the secondary index is very hard to determine which primary index it goes with. Whenever you use asterisks to indicate statistical significance, you should put under the table what the asterisk means
Table 8, page 13, one line in Individual fixed effects shows as Chinese characters!
6. Figures are fine.
7. Edits:
Line 31 first page: the reference shows as “Reference source not found”
Line 289-291, a phrase repeats. Please correct.
Line 310: “…is more obvious, and…” should read “…more obvious, with”
Comments on the Quality of English Language
See above. English grammar is well done.
Author Response
We are very grateful to thank your good evaluation and constructive suggestions, which have significantly improved the clarity of this paper. We have revised our paper based on your kind advice. All revisions are in a red color in the revised paper.Please refer to the attachment for details.
Brief response:
- According to your suggestion, we have italicized the special nouns that appear only once or twice in the article
- We have expanded the data source and evaluation indicator System section to provide more information about the original data set (lines 96-111). The process of handling outliers is supplemented. In addition, we will provide more details about the sample screening process to explain why the final sample size is 4090, and discuss the representation of these participants in the overall elderly population or China, in fact, the dataset is an open dataset, we found through the matching of ids that 4090 covers all regions of China.
- We put too much emphasis on the impact on social support and income and have revised this section
- We have embellished the form
- We have corrected the spelling mistakes and phrase errors in the article
Thanks again for your advice!

Reviewer 4 Report
Comments and Suggestions for Authors
The basic theme of the paper is an interesting one, but there are some major issues to be solved, along with minor ones.
The references are very few and need to be significantly improved and enriched. This will also help improving the Literature review section which is very summary, too.
The authors should explain more clearly how their index, which is based on a previous index specific exclusively for smart cities, can be applied for smart regions, while some indicators as those for Skilled Human capital and also some others are specifically targeting only urban areas. This could affect the results even if the authors use some control variables.
Even if the idea and the models are interesting, how reliable are the results of an OLS model based on the data from only 2 years?
This paper needs a review to correct English syntax, starting from the Abstract. There are also errors regarding the References (page 1).
The minor issues regard especially some of the tables, as follows:
Table 1 should separate clearly the Indexes accordingly to the risks. As it is now some Indexes may be perceived as linked to different risks than the ones intended. Moreover, it is impossible for any other than the authors to confirm or infirm the value of Weight Coefficients.
Table 2 has a similar problem with Table 1, making difficult to follow the correspondence of the data between the columns.
Table 8 includes Chinese characters which are different than the rest of the paper.
Comments on the Quality of English LanguageThe basic theme of the paper is an interesting one, but there are some major issues to be solved, along with minor ones.
The references are very few and need to be significantly improved and enriched. This will also help improving the Literature review section which is very summary, too.
The authors should explain more clearly how their index, which is based on a previous index specific exclusively for smart cities, can be applied for smart regions, while some indicators as those for Skilled Human capital and also some others are specifically targeting only urban areas. This could affect the results even if the authors use some control variables.
Even if the idea and the models are interesting, how reliable are the results of an OLS model based on the data from only 2 years?
This paper needs a review to correct English syntax, starting from the Abstract. There are also errors regarding the References (page 1).
The minor issues regard especially some of the tables, as follows:
Table 1 should separate clearly the Indexes accordingly to the risks. As it is now some Indexes may be perceived as linked to different risks than the ones intended. Moreover, it is impossible for any other than the authors to confirm or infirm the value of Weight Coefficients.
Table 2 has a similar problem with Table 1, making difficult to follow the correspondence of the data between the columns.
Table 8 includes Chinese characters which are different than the rest of the paper.
Author Response
Thank you very much for your insightful and valuable comments and constructive suggestions, which have significantly improved the clarity of this paper. We have revised our paper based on your kind advice. All revisions are in a red color in the revised paper.Please refer to the attachment for details.
Brief response:
- According to your suggestion, it is true that our article lacks documentary evidence in many places, so we have expanded the literature
- We explain the indicators in the indicator system. For the indicators only targeting urban areas, we mainly assign low weights in the paper to minimize the impact
- The two-year data does have some limitations, but considering that the survey subjects are the elderly, the sample size will be greatly reduced if the research time span is too large
- We have embellished the table and corrected the incorrect spelling and phrases in the article
Thanks again for your advice!

Reviewer 5 Report
Comments and Suggestions for Authors
Having read the article, we wonder if not a few terms seem to exist as awkward translations of non-English constructs that needed more clarification. The abstract, for instance, refers to a vague notion of "cleverness" that does not go through further refinement for the rest of the article; as it turns out, the actual word "cleverness" never actually appears after the abstract.
The introduction also seems to assume an overly high amount of preexisting knowledge on the part of the audience. Among other notions mentioned in the section, the terms "precise poverty alleviation" (line 41) and "old age allowance" (line 41) almost assuredly require some more elaboration and references to secondary literature and/or government memoranda and policy papers, given the preeminence of the difficult issues that the government must face in managing the concerns of the elderly.
In the literature review and research hypothesis section, the authors should have had some sort of transition between the paragraph on research regarding the hazards of poverty in old age (lines 50~65) with the paragraph on the need for regional smart zones (lines 66~83); the connection between the two paragraphs does not seem apparent at first glance. We would have also benefited from a clearer explanation on the difficulties regarding the elimination of information barriers between regions (lines 72~73) and the somewhat hackneyed notion of the so-called unique traditional culture of the nation (lines 76~77); the argumentative nature of such statements necessitate some sort of elaboration grounded in the secondary literature.
In section 3.3, the authors refer to the moderating effects of family endowment and social capital on the dimensions of poverty risk in old age (lines 194~197), but again, we would have appreciated more effort in trying to more fully explain a point that might seem obvious to the authors, but not immediately apparent to the audience. In the first half of section 3.4 (lines 244~252), one has a feeling that the authors simply listed some ideas referenced in the secondary literature, but we would have appreciated more argumentative insight from the authors themselves; in other words, the authors should show more argumentative proclivities in justifying the inclusion of section 3.4.
Lines 276 through 277 specifically refer to the lower development level of the western regions of China, and the authors highlight this basic idea again in the research conclusions (lines 391~393). On the other hand, China's notoriously less well developed western regions (relative to the eastern regions) deserve a fuller introductory treatment for readers who might not immediately understand this foundational insight on geographic disparities in China's development. Of course, the authors use plenty of statistical insight and research results to testify to this reality, but the foundational insight required on the weak development of Western China depends not only on the presentation of research results, but also on interdisciplinary literature (texts relating to history, anthropology, and agriculture/environmental studies come to mind) on the topic.
Line 406 specifically refers to the impact of gender on the vulnerability of the elderly who grapple with he perils of poverty, but the body of the paper has only a smattering of references to gender (this reviewer counts literally four instances of the word "gender" in the main body of the manuscript). In other words, it seems that the authors have presented conclusions on gender that the authors have not actually earned the right to make. Again, to rectify this issue, the authors should probably have a new body section on the gender-based disparities of poverty that can fall upon elderly people who already suffer from diminished access to smart cities/localities.
In short, the authors of the paper would need to expand on the scholarly literature in terms of treating the topic of this paper (i.e., by more fully elaborating on interdisciplinary sources that touch on disciplines such as history, anthropology, and environmental studies), work on presenting materials that seem more argumentative (rather than merely presenting the findings and insights of other authors), and clarify certain ideas that seem highly fascinating but require more elaboration.
Author Response
Thank you very much for your insightful and valuable comments and constructive suggestions, which have significantly improved the clarity of this paper. We have revised our paper based on your kind advice. All revisions are in a red color in the revised paper.Please refer to the attachment for details.
Brief response:
- We have revised the nouns that appear only once in the abstract, and corrected the incorrect spelling and phrases in the article
- According to your suggestions, we have explained the concepts in the article, such as targeted poverty alleviation, in detail by providing annotations and expanding the documentary evidence
- In the literature review part, we added the transition section to form a better logic
- In Section 3.3 we add the argument for the moderating effects of social support and family endowments, and in section 3.4 we add our own summary
- We have added an introduction to the Western region (lines 301-309)
- The article put too much emphasis on gender content, and we have revised it
Thanks again for your advice!

Round 2
Reviewer 3 Report
Comments and Suggestions for Authors
All prior concerns have been addressed making this manuscript flow well, logical and with excellent impact. Thank you for the opportunity to review this manuscript again.
Author Response
Thank you for your review and approval!
Reviewer 4 Report
Comments and Suggestions for Authors
I consider that authors remediated the issues pointed by me.
Author Response
Thank you for your review and approval!
Reviewer 5 Report
Comments and Suggestions for Authors
The authors of the paper largely incorporated the spirit of the suggested revisions, and this paper will probably form a beginning of studies related to poverty in the somewhat understudied Western regions of mainland China.
Author Response
Thank you for your review and approval!